

**Developing fragility functions for aquaculture rafts and eelgrass in the case of the 2011 Great**
**East Japan tsunami**
Anawat SUPPASRI[1], Kentaro FUKUI[2], Kei YAMASHITA[3], Natt LEELAWAT[4], Hiroyuki
OHIRA[5], and Fumihiko IMAMURA[6]
[1]International Research Institute of Disaster Science, Tohoku University
(468-1 Aramaki-aza Aoba, Aoba-ku, Sendai 980-0845, Japan) suppasri@irides.tohoku.ac.jp
[2]Kanagawa Prefectural Office,
(1 Nihon Odori, Naka-ku, Yokohama 231-8588, Japan) fukui19940616@gmail.com
[3]International Research Institute of Disaster Science, Tohoku University
(468-1 Aramaki-aza Aoba, Aoba-ku, Sendai 980-0845, Japan) yamashita@irides.tohoku.ac.jp
[4] Department of Industrial Engineering, Faculty of Engineering, Chulalongkorn University
(Phayathai Road, Pathumwan, Bangkok 10330 Thailand) natt.l@chula.ac.th
[5] Electric Power Development Co., Ltd.
(6-15-1,Ginza, Chuo-ku, Tokyo,104-8165 Japan) Hiroyuki_Oohira@jpower.co.jp
[6]International Research Institute of Disaster Science, Tohoku University
(468-1 Aramaki-aza Aoba, Aoba-ku, Sendai 980-0845, Japan) imamura@irides.tohoku.ac.jp

**Abstract**
Since the two devastating tsunamis in 2004 (Indian Ocean) and 2011 (Great East Japan), new
findings have emerged on the relationship between tsunami characteristics and damage in terms
of fragility functions. Human loss and damage to buildings and infrastructures are the primary
target of recovery and reconstruction; thus, such relationships for offshore properties and marine
ecosystems remain unclear. To overcome this lack of knowledge, this study used the available data
from two possible target areas (Mangokuura Lake and Matsushima Bay) from the 2011 Japan
tsunami. This study has three main components: 1) reproduction of the 2011 tsunami, 2) damage
investigation and 3) fragility function development. First, the source models of the 2011 tsunami
were verified and adjusted to reproduce the tsunami characteristics in the target areas. Second, the
damage ratio of the aquaculture raft and eelgrass was investigated using satellite images taken
before and after the 2011 tsunami through visual inspection and binarization. Third, the tsunami
fragility functions were developed using the relationship between the simulated tsunami
characteristics and the estimated damage ratio. Based on the statistical analysis results, fragility
functions were developed for Mangokuura Lake, and the flow velocity was the main contributor
to the damage instead of the wave amplitude. For example, the damage ratio above 0.9 was found
to be equal to the maximum flow velocities of 1.3 m/s (aquaculture raft) and 3.0 m/s (eelgrass).
This finding is consistent with the previously proposed damage criterion of 1 m/s for the
aquaculture raft. This study is the first step in the development of damage assessment and planning
for marine products and environmental factors to mitigate the effects of future tsunamis.
**Keywords**: 2011 Great East Japan tsunami, fragility functions, aquaculture raft, eelgrass



## 1. Introduction

Aquaculture and ecological systems provide many services and functions to humans and are important to the global economy (Costanza et al., 1997). The 2011 Great East Japan tsunami caused devastating damage to inland and offshore properties. Considerable economic damage from the loss of aquaculture products and the impact to ecological systems was also caused by this tsunami. Since the 2004 Indian Ocean tsunami and the 2011 tsunami, numerous quantitative measures of tsunami vulnerability, such as fragility functions, have been developed for buildings (Suppasri et al., 2016), infrastructures (Shoji and Nakamura, 2017) and marine vessels (Suppasri et al., 2014 and Muhari et al., 2015). However, only one criterion is based on a previous study of the 1960 Chilean tsunami that struck the west of Japan: the damage to an aquaculture raft (pearl) begins to occur when the tsunami flow velocity is larger than 1 m/s regardless of the water level (Nagano et al., 1991). No other criterion or study has been presented regarding the vulnerability of marine plants.

### 1.1 Objectives

To quantitatively assess such damage to marine products and marine ecosystems, the main objective of this study is to develop the fragility functions as the first step to understand the relationship between the tsunami characteristics and the damage. After reviewing previous works, this study comprises three main sections: 1) reproduction of the 2011 tsunami, 2) damage investigation and 3) development of fragility functions. The first section presents a validation of the proposed source models for the 2011 tsunami and the adjustment for tsunami reproduction in the study areas. The second section presents the available damage data and damage quantification. The third section presents statistical analysis methods to develop the fragility functions using the results obtained from the first and second sections. Finally, new findings, recommendations and the limitations of this study are discussed.

### 1.2 Review of previous studies

This section reviews selected previous studies related to the damage characteristics of offshore facilities and marine plants against tsunamis. The first attempt was based on the 1960 Chilean tsunami that struck the west of Japan. The damaged aquaculture rafts were plotted against the simulated maximum water level and flow velocity (Nagano et al., 1991). As shown in Fig. 1, the damage to the aquaculture raft (pearl) begins to occur when the tsunami flow velocity is higher than 1 m/s regardless of the water level. Similarly, Kato et al., (2010) applied identical criteria to quantify the damage to aquaculture rafts in areas along the east coast of Japan, which were struck by the 2010 Chilean tsunami. They found that the damage on the east coast of Japan caused by the 2010 Chilean tsunami was accurately modeled by the proposed damage criteria developed from the data of the 1960 Chilean tsunami in the west of Japan.

After the 2011 tsunami, Suppasri et al. (2014) and Muhari et al. (2015) developed fragility functions for fishing boats. Based on their results, the threshold water level and flow velocity values for the complete destruction of small boats of less than 5 tons are 2 m and 1 m/s, respectively. Keen et al. (2017) developed fragility functions for structural components in small craft harbors based on actual damage caused by the 2011 tsunami on the US west coast. The 2016 Fukushima tsunami caused no inland damage but some damage to aquaculture rafts and fishing boats in Sendai Bay (Suppasri et al., 2017). Nevertheless, no damage criteria or fragility functions have been proposed for the 2011 tsunami. There have been limited studies on the relation between tsunami




characteristics and damage to sea plants. Sakamaki et al. (2016) and Tsujimoto et al. (2016)
reported the damage to eelgrass in Matsushima Bay but provided no direct consideration of the
effect of tsunami characteristics. Yamashita et al. (2016) noted possible relationships between the
sediment deposition and erosion caused by the 2011 tsunami and the damage to eelgrass.

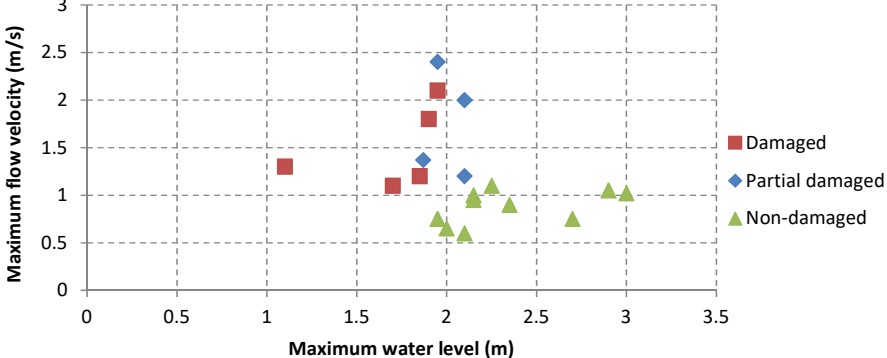

Fig. 1 Damage criteria of the aquaculture raft based on the damage data from Kii Peninsula,
western Japan, from the 1960 Chilean tsunami (Adapted from Nagano et al., 1991)

**1.3 Target areas of this study**
Because the size of the 2011 tsunami was extremely large, most aquaculture rafts and other marine
plants were completely destroyed. There are only two well-suited locations with specific coastal
geography, namely, Mangokuura Lake and Matsushima Bay in Miyagi Prefecture (Fig. 2), where
the effects of the tsunami were comparatively small (Suppasri et al., 2012) and the aquaculture
rafts were undamaged and the eelgrass survived (University of Tokyo, 2016). Mangokuura Lake
has a notably narrow entrance from the Pacific Ocean through Ishinomaki Bay, and the average
sea depth is as shallow as 5 m or less. Matsushima Bay is protected by almost 300 small islands
around the bay front. Thus, the 2011 tsunami inundation and run-up heights in both areas were less
than 1-2 m, whereas they were as high as 10 m in other nearby areas (Suppasri et al., 2012). As a
result, some aquaculture rafts and other marine plants survived in these two locations, which
enabled the development of fragility functions.



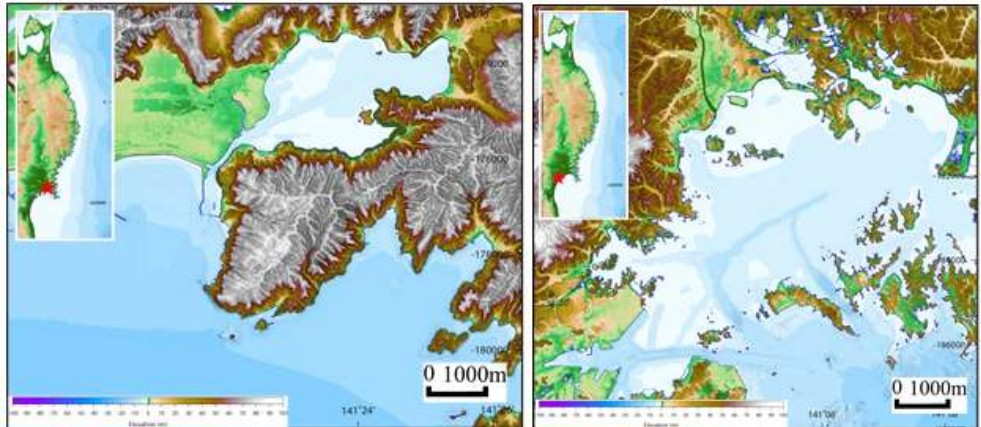

Fig. 2 Study areas: (a) Mangokuura Lake and (b) Matsushima Bay

**2. Reproduction of the 2011 tsunami**
**2.1 Simulation conditions**
To obtain tsunami-related parameters, including the water level and flow velocity, the 2011
tsunami was reproduced using a numerical analysis. The 2011 tsunami was numerically simulated
using a set of nonlinear shallow water equations, which were discretized using the staggered leap-
frog finite difference scheme (TUNAMI model) with bottom friction in the form of Manning's
formula, similar to previous studies (Suppasri et al., 2010, Charvet et al., 2015 and Macabuag et
al., 2016). Six computational domains were used as a nesting grid system of 1,215 m (Region 1),
405 m (Region 2), 135 m (Region 3), 45 m (Region 4), 15 m (Region 5) and 5 m (Region 6). The
tidal level of –0.42 m was set at the time of the tsunami occurrence, and the simulation time was
set to three hours to maximize the water level and flow velocity.



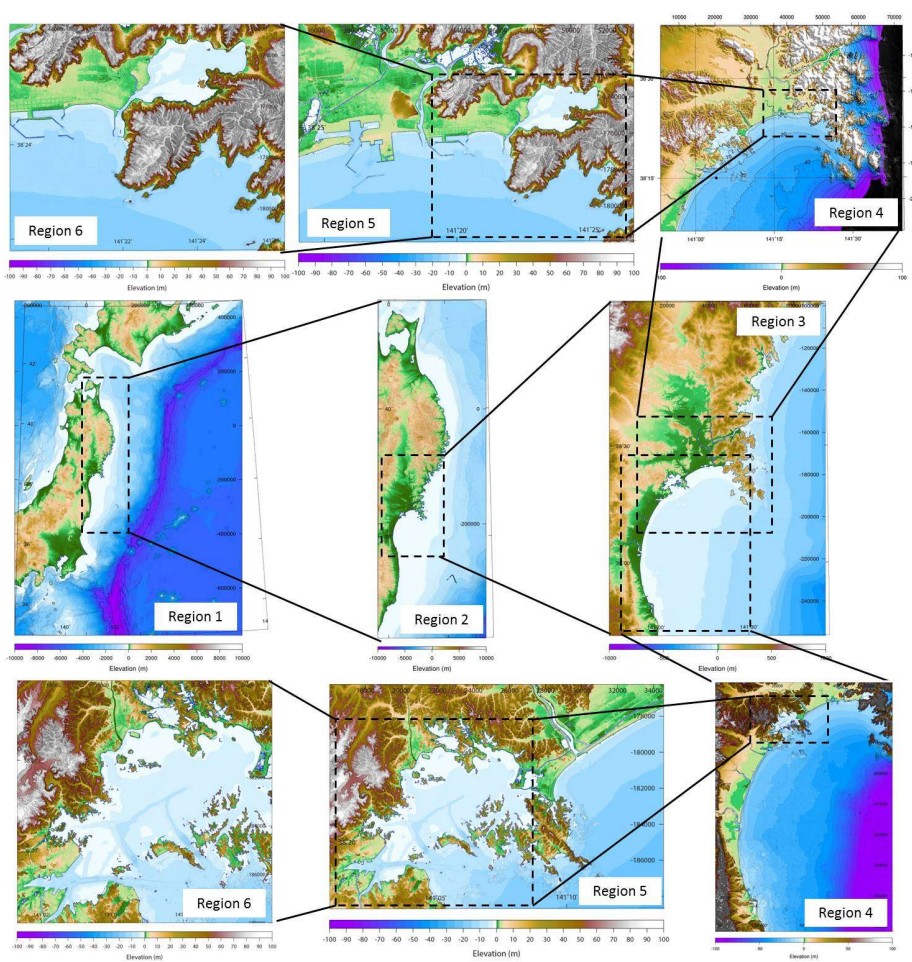

Fig. 3 Six computational areas for Mangokuura Lake (up) and Matsushima Bay (down)

**2.2 Model calibration and verification**

Three models of fault parameters were selected to reproduce the 2011 tsunami: Model 1: Tohoku
University model (Imamura et al., 2013); Model 2: Satake model (Satake et al., 2013); and Model
3: Japan Nuclear Energy Safety Organization (JNES) model (Sugino et al., 2013). The
corresponding fault parameters were used to estimate the seafloor deformation proposed by Okada
(1985), which later became the initial seafloor condition for the tsunami numerical simulation. The
simulated tsunami inundation and run-up height with the actual measured values (Mori et al., 2012)
were validated for each area using Aida's $K$ and $\kappa$ (Aida, 1978) as defined below.



$$\log K = \frac{1}{n}\sum_{i=1}^{n}\log K_i \tag{1}$$

$$\log\kappa = \sqrt{\frac{1}{n}\sum_{i=1}^{n}(\log K_i)^2 - (\log K)^2} \tag{2}$$

$$K_i = \frac{x_i}{y_i} \tag{3}$$

where $x_i$ and $y_i$ are the measured and simulated tsunami trace heights, respectively, at point $i$. Consequently, $K$ is considered a correction factor to adjust the modeled values to fit the actual tsunami averaged over several locations; $\kappa$ is defined as a measure of the fluctuation or deviation in $K_i$. The values of Aida's $K$ and $\kappa$ from each model are shown in Table 1.

For Mangokuura Lake, Model 3 produced the optimal values of Aida's $K$ and $\kappa$. Because $K$ is slightly less than 1.0, the simulated tsunami heights are slightly larger than the measurement. Similarly, for Matsushima Bay, Model 2 produced the best Aida`s $K$ and $\kappa$. Because $K$ is larger than 1.0, the simulated tsunami heights are smaller than the measurement. To better obtain the tsunami parameters, the fault slip was scaled by the $K$ values of 0.96 and 1.29 for Mangokuura Lake and Matsushima Bay, respectively, so that the reproduced tsunami closely matched the measured tsunami trace heights and satisfied the guideline of the Japan Society of Civil Engineers; $0.95 < K < 1.05$ and $\kappa < 1.45$ (Suppasri et al., 2010). As a result, the accuracy of the simulated tsunami parameters in both study areas was improved, as shown in Fig. 4.

**Table 1** Aida's $K$ and $\kappa$ for each model and after the model scaling

| Location | Value | Model 1 | Model 2 | Model 3 | After scaling (Model 2) | After scaling (Model 3) |
|---|---|---|---|---|---|---|
| Mangokuura Lake | $K$ | 0.90 | 0.87 | 0.96 | - | 1.01 |
| | $\kappa$ | 1.65 | 1.49 | 1.45 | - | 1.41 |
| Matsushima Bay | $K$ | 1.53 | 1.29 | 1.35 | 1.06 | - |
| | $\kappa$ | 1.45 | 1.34 | 1.42 | 1.39 | - |



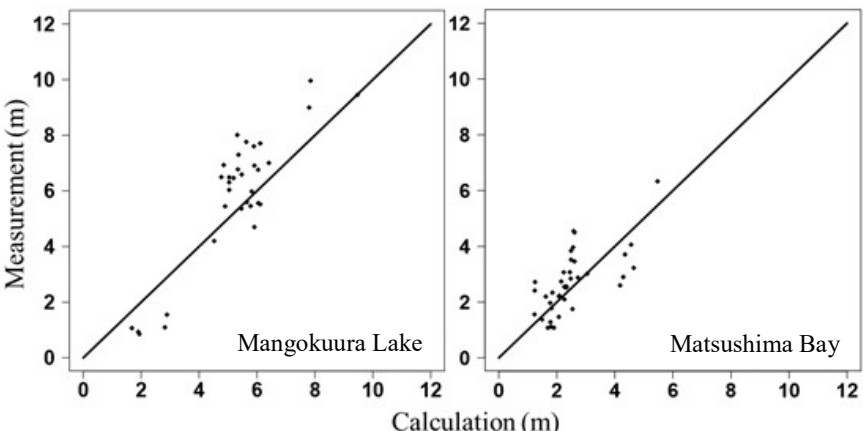

Fig. 4 Comparison of the simulated and measured tsunami heights in Mangokuura Lake and
Matsushima Bay

**2.3 Reproduction results**
The hydrodynamic properties of the 2011 tsunami were reproduced based on the model calibration
and verification as mentioned above. Fig. 5 shows that the average maximum water level and flow
velocity in the bay of Mangokuura Lake are approximately 0.5 m and 1-2 m/s, those of Matsushima
Bay are approximately 2 m and 3-5 m/s, and the average offshore maximum water level and flow
velocity in the other 2011 tsunami affected areas were much higher than these values (Suppasri et
al., 2014).

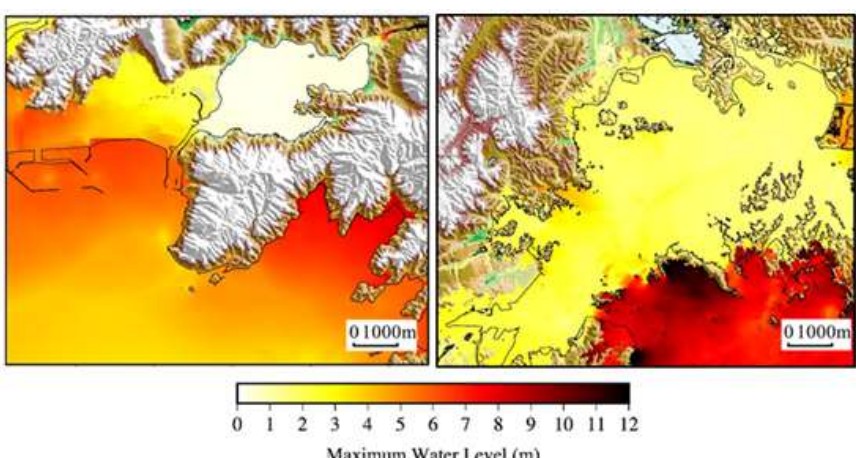






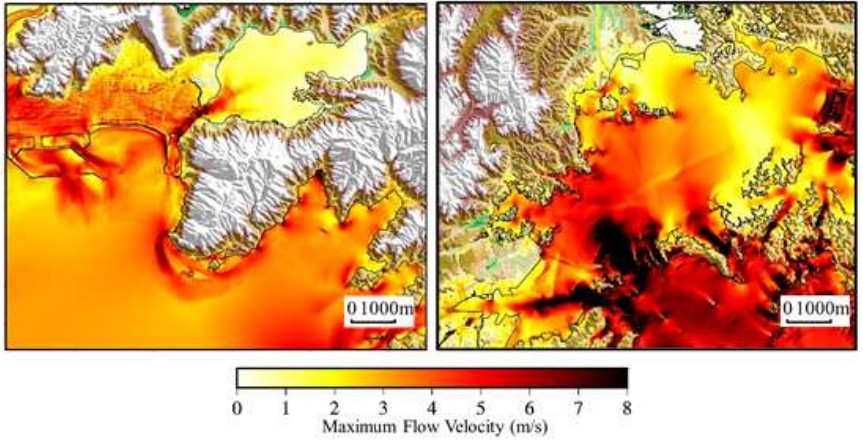

Fig. 5 Simulated maximum water level and flow velocity in Mangokuura Lake and Matsushima
Bay

## 3. Damage investigation of the aquaculture rafts and eelgrass

Damage inspection was performed using satellite images taken before and after the tsunami
through a visual inspection for the aquaculture rafts and an image analysis for the eelgrass.

### 3.1 Damage investigation of the aquaculture rafts

In this study, only the long-line type of aquaculture raft (Fig. 6) had sufficient quantities to develop
the fragility function. This type of aquaculture raft is common in the study area and is used for
oyster and seaweed farming. Examples of the visual inspection of the aquaculture rafts in the lake
before (Fig. 7a) and after the tsunami (Fig. 7b) are shown. Approximately half of the rafts remained
after the tsunami; the others were completely washed away. The remaining aquaculture rafts were
classified as undamaged, whereas the disappeared aquaculture rafts were classified as damaged.
Fig. 7 also shows the visual inspection results as polygons of the undamaged and washed-away
aquaculture rafts (long-line type) in Mangokuura Lake. Many damaged aquaculture rafts were
found near the entrance to and in the middle of the lake. Then, the created polygons were gridded
into $5 \times 5$ m$^2$ regions corresponding to the finest tsunami simulation grid (Region 6). The simulated
maximum water level and flow velocity were assigned to each grid. For Matsushima Bay, there
was an insufficient number of long-line-type aquaculture rafts, and many rafts could not be
classified into types. Therefore, only damaged aquaculture rafts in Mangokuura Lake were used
to develop fragility functions.



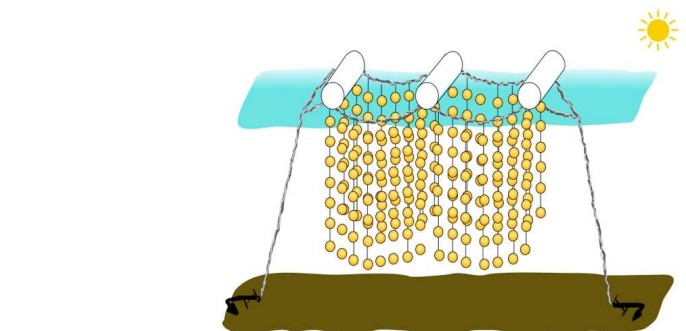

Fig. 6 Aquaculture raft (long-line type)

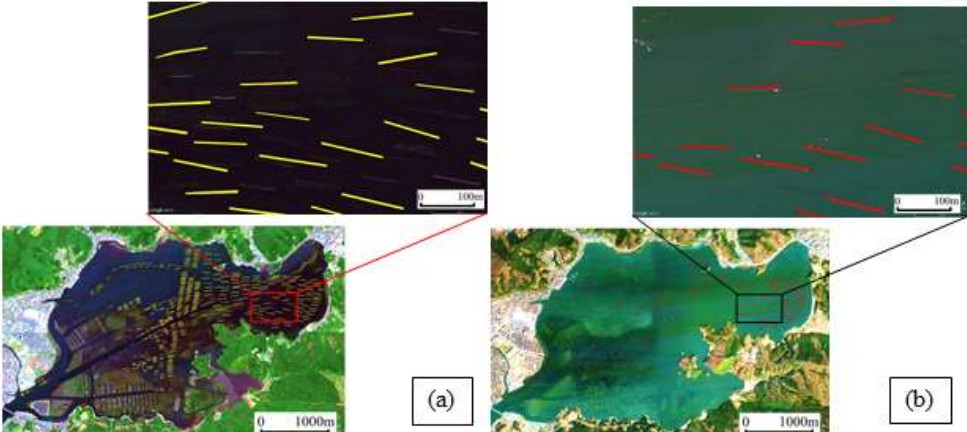

Fig. 7 Visual damage interpretation of aquaculture rafts (long-line type) (a) before and (b) after
the 2011 tsunami

**3.2 Damage investigation of eelgrass**

Damage to eelgrass occurs in one of three modes: cut-off, deposition or erosion, as shown in Fig. 8. Although the deposition and erosion can be estimated using a sediment transport model, more detailed data and surveys are required to obtain the necessary data for the model input. This pilot study considered only the tsunami itself. In addition, the erosion was controlled primarily by the flow velocity. Therefore, the cut-off and erosion were considered damage from the horizontal force of the tsunami.

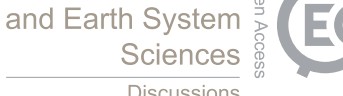



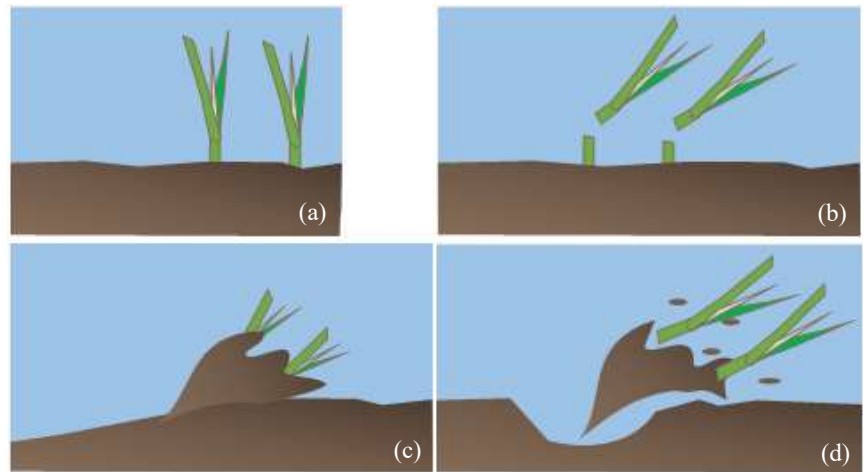

Fig. 8 Eelgrass (a) and its damage pattern: (b) cut-off, (c) sand deposition and (d) erosion

Color images from the actual satellite image before the 2011 tsunami and after the 2011 tsunami were analyzed (University of Tokyo, 2016 and Tsujimoto et al., 2016). At this stage, the areas for land, sea, aquaculture raft, eelgrass and mudflat were first identified. To identify only the eelgrass area, the colored images were binarized to binary (black and white) images using the image analysis software ImageJ which is being developed at the National Institutes of Health, the United States (ImageJ, 2016). This binarization helps distinguish eelgrass and non-eelgrass areas. Figs. 9 and 10 show the eelgrass areas before and after the 2011 tsunami in Mangokuura Lake and Matsushima Bay, respectively. The identified damage and undamaged areas for both aquaculture rafts and eelgrass were gridded into $5\times5$ $m^2$ regions. Then, the damage ratio of each grid was calculated, and the maximum simulated water level and flow velocity were assigned to each grid. Finally, another process was performed to create a list of the simulated tsunami characteristics (water level and velocity) and damage ratio to develop the fragility function, as explained in the next section.

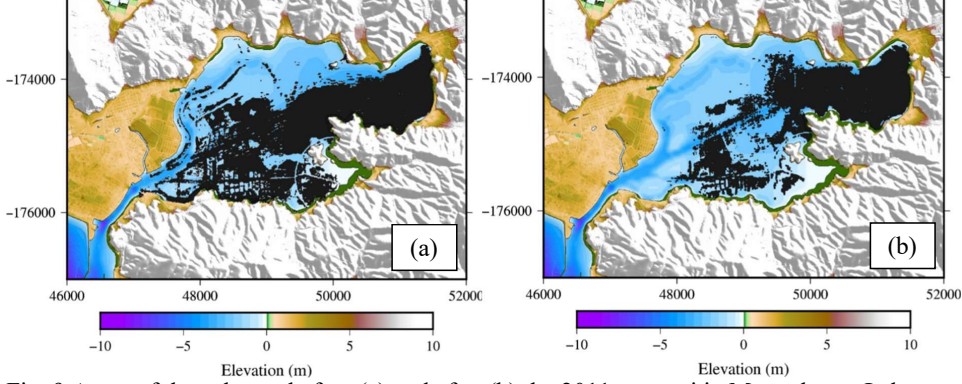

Fig. 9 Areas of the eelgrass before (a) and after (b) the 2011 tsunami in Mangokuura Lake

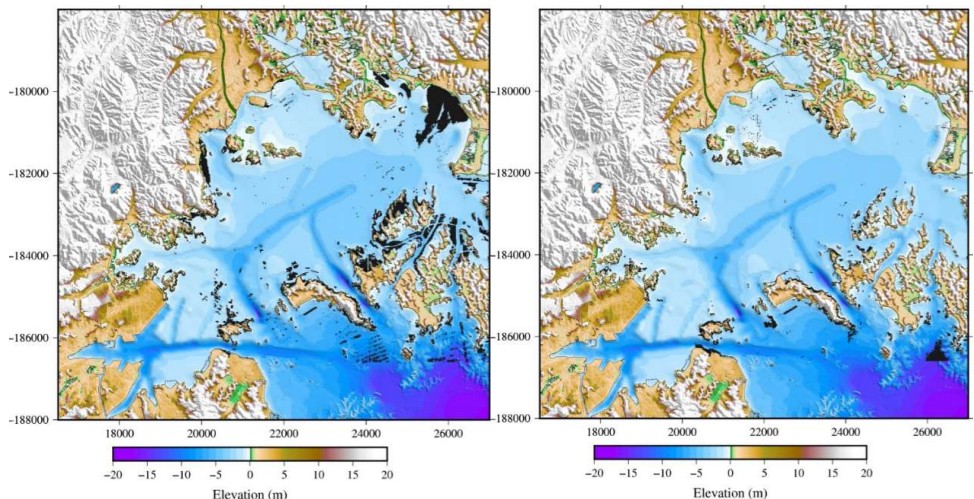

Fig. 10 Areas of the eelgrass before (a) and after (b) the 2011 tsunami in Matsushima Bay

**4.  Developing tsunami fragility functions**
**4.1 Preliminary analysis**
A comparison of the aquaculture raft data in the cases of the 1960 Chilean tsunami (Fig. 1) and
the 2011 Japan tsunami is shown in Fig. 11. Most of the undamaged aquaculture rafts in the 2011
tsunami were limited to the maximum flow velocity less than 1.5 m/s. For both target areas, the
damage probabilities for each range of the simulated water level and maximum flow velocity of
both aquaculture rafts and eelgrass were calculated and are shown against a median value in a
specific range of the grids. In Fig. 12, the preliminary scatter plot does not show any significant
trend between the simulated maximum water level and the damage to the aquaculture rafts (Fig.
12a) and eelgrass (Fig. 12b) in Mangokuura Lake or between the simulated maximum flow
velocity and the damage to eelgrass in Matsushima Bay (Fig. 12c). Thus, another expected
parameter was used to develop the fragility functions: the simulated maximum flow velocity in
Mangokuura Lake. To verify that our regression model is better than the predicted average value,
an analysis of variance (ANOVA) was performed. The ANOVA is a statistical test to verify
whether the regression model is significantly satisfactory in terms of predicting the variable's value.
The analysis can test whether the proposed regression model provides a better estimation than
using the average value of the predicted variables. The result shows that the calculated models
significantly predict the damage ratio ($F$ aquaculture raft = 74.73; $p$ aquaculture raft < 0.001; $F$
eelgrass = 89.70; $p$ eelgrass < 0.001) in the model.



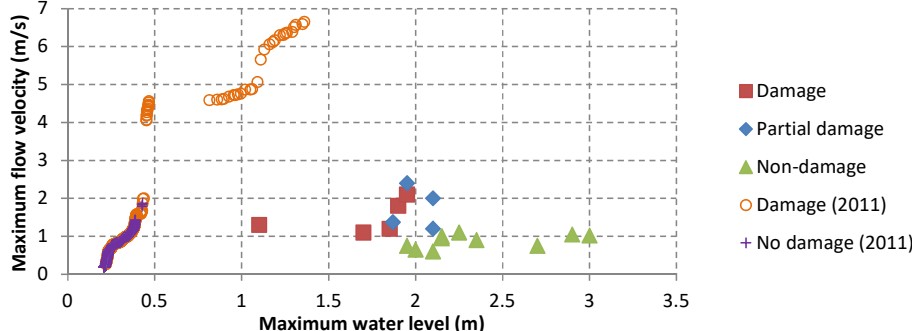

Fig. 11 Comparison of the aquaculture raft data from the 1960 Chilean tsunami (Fig. 1) and the
present study on the 2011 Japan tsunami

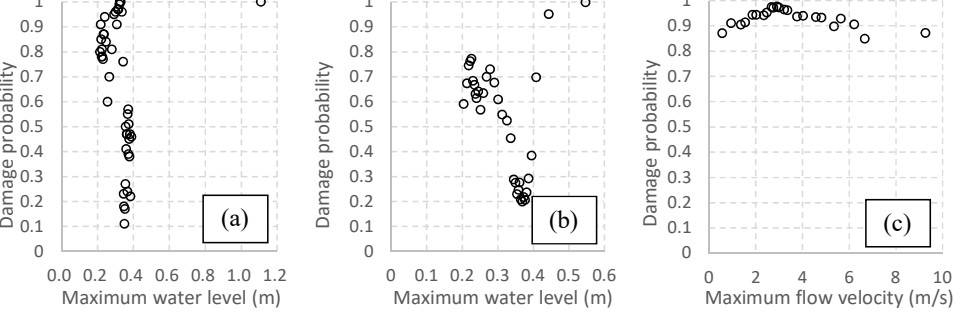

Fig. 12 Maximum water level and damage probability of the (a) aquaculture rafts and (b) eelgrass
in Mangokuura Lake and (c) eelgrass in Matsushima Bay

**4.2 Linear regression analysis**
Only the simulated maximum flow velocity and damaged-eelgrass data in Mangokuura Lake could
be used to develop the fragility functions. The tsunami fragility functions were developed by
applying the classical standardized lognormal distribution function throughout the linear
regression analysis for both aquaculture rafts and eelgrass. For Mangokuura Lake, Fig. 12 shows
the histograms of the numbers of damaged and undamaged aquaculture rafts in every 100 grids
(Fig. 13a) and 0-50% damaged and 50-100% damaged eelgrass in every 5,000 grids (Fig. 13b) in
terms of the simulated maximum flow velocity range. Both histograms show that the damage data
increase when the flow velocity increases. A linear regression analysis was performed to develop
the fragility function. The cumulative probability $P$ of occurrence of the damage is given in Eq.
(4).

$$P(x) = \Phi\left[\frac{\ln x - \mu'}{\sigma'}\right] \qquad (4)$$




where $\Phi$ is the standardized lognormal distribution function, $x$ is the hydrodynamic feature of the
tsunami (simulated maximum velocity), and $\mu'$ and $\sigma'$ are the mean and standard deviation of ln x,
respectively. The statistical parameters $\mu'$ and $\sigma'$ of the fragility function were obtained by plotting
ln $x$ against the inverse of $\Phi^{-1}$ on lognormal probability papers and performing least-squares fitting
of this plot (Figs. 14a and 14b). Consequently, two parameters are obtained as the intercept (= $\mu'$)
and angular coefficient (= $\sigma'$) in Eq. (5).
$$\ln x = \sigma'^{\Phi^{-1}} + \mu' \qquad\qquad (5)$$

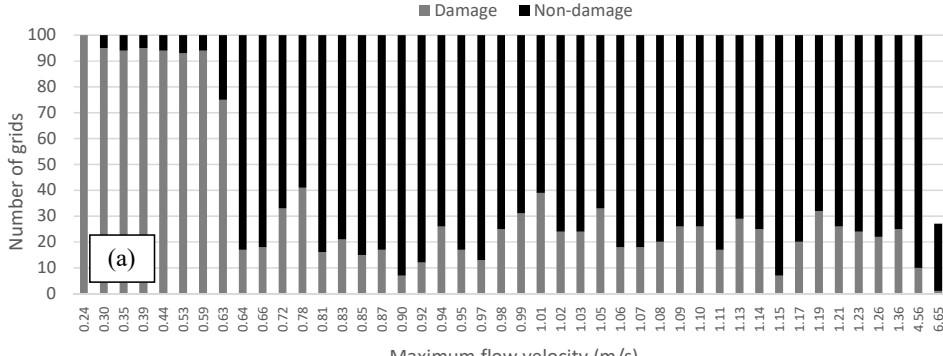


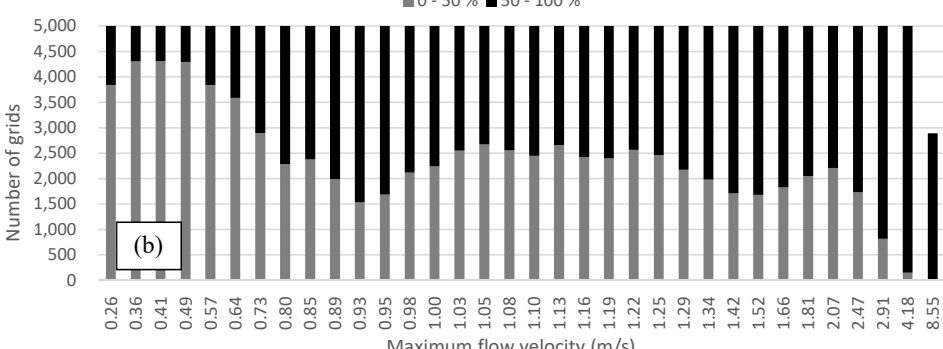

Fig. 13 Histogram of the numbers of (a) damaged and undamaged aquaculture rafts and (b) 0-50%
damaged and 50-100% damaged eelgrass in terms of the simulated flow velocity range in
Mangokuura Lake.




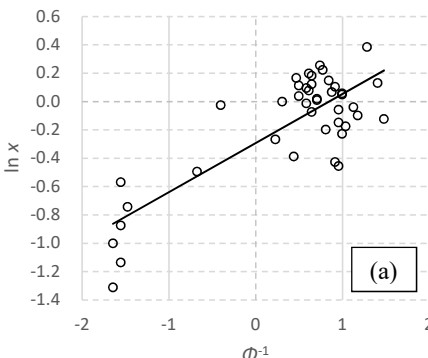 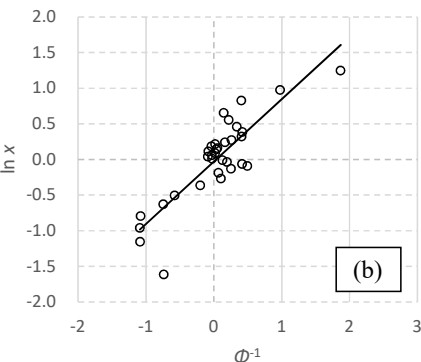

Fig. 14 Least-squares fit on lognormal probability paper for the aquaculture rafts (a) and eelgrass (b) in Mangokuura Lake

**4.3 Tsunami fragility functions for the aquaculture rafts and eelgrass**

With the regression analysis, the parameters that best fit the fragility functions with respect to the maximum flow velocity are shown in Table 2. The tsunami fragility curves for the aquaculture rafts and eelgrass were developed as shown in Figs. 15a and 15b, respectively. The proposed fragility functions show that a damage ratio above 0.5 corresponds to the maximum flow velocity of 0.8 m/s (aquaculture raft) and 1.0 m/s (eelgrass). A damage ratio above 0.9 corresponds to the maximum flow velocity of 1.3 m/s (aquaculture raft) and 3.0 m/s (eelgrass). The results for the aquaculture rafts are consistent with the previously proposed criteria (Nagano et al., 1991): at 1 m/s flow velocity, the damage ratio is almost 0.8.

**Table 2** Parameters to create the tsunami fragility functions.

| Item | $\mu'$ | $\sigma'$ | $R^2$ |
|---|---|---|---|
| Aquaculture raft | -0.2917 | 0.3464 | 0.65 |
| Eelgrass | -0.0314 | 0.8750 | 0.74 |





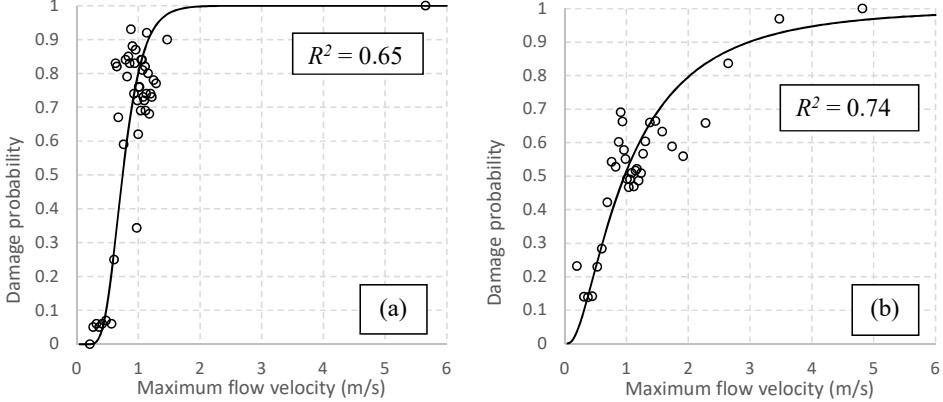

Fig. 15 Tsunami fragility functions for the aquaculture rafts (a) and eelgrass (b) based on data from Mangokuura Lake

## 5. Conclusions
### 5.1 Main findings

This study was the first attempt in this field to develop fragility functions for aquaculture rafts and eelgrass. The careful selection of the study areas and availability of the damage data enabled this attempt. First, we reproduced the hydrodynamic characteristics, i.e., the water level and flow velocity of the 2011 tsunami, using the tsunami trace data for the model calibration and verification based on the finest grid of $5\times5$ m$^2$ regions. The damage data for both aquaculture rafts and eelgrass were investigated by visually inspecting and analyzing the satellite images before and after the 2011 tsunami. Then, the fragility functions for the aquaculture rafts and eelgrass were developed using the data for Mangokuura Lake. This lake appears to be the only suitable location for a study based on tsunami characteristics because of its location and consequent damage range from no damage to little damage to considerable damage. In addition, Matsushima Bay was exposed to a stronger tsunami and had fewer undamaged aquaculture rafts and surviving eelgrass. The main conclusions are as follows:

- Based on the reproduced hydrodynamic characteristics of the 2011 tsunami, Matsushima Bay was hit by a stronger tsunami than Mangokuura Bay (Fig. 5).
- The maximum water level is not related to the damage to aquaculture rafts and eelgrass (Fig. 12).
- The threshold value (at 90% damage probability) of the maximum flow velocity for damage to aquaculture rafts and eelgrass is 1.3 m/s and 3.0 m/s, respectively (Fig. 15).
- The proposed fragility function for the aquaculture rafts is consistent with the previously proposed damage criteria and can further provide the values of the damage ratio at other flow velocities in addition to the threshold value.
- This information on the tsunami damage in offshore areas is expected to be useful for marine product and environmental damage assessment and recommendations for aquaculture raft zoning to mitigate the effects of tsunamis in the future.





**5.2 Limitations, considerations and future studies**
Although this study successfully developed fragility functions for aquaculture rafts and eelgrass
for the first time, certain limitations and considerations exist when applying the fragility functions,
and possible improvements to be pursued in future studies are as follows.
- The developed fragility functions may underestimate the economic damage related to
aquaculture rafts because the loss of marine products may occur even when the rafts remain.
For example, although the aquaculture rafts were present in the satellite image, in some cases,
the marine products were completely washed away or damaged when the rafts collided with
each other.
- This study simulated only the hydrodynamic characteristics of the tsunami, which can directly
explain the damage caused by cut-off and erosion. However, the damage caused by deposition
was not considered.
- The use of the actual surveyed damage to the aquaculture rafts and eelgrass and the application
of a sediment transport model may increase the accuracy of the fragility functions.
- The fragility functions for both aquaculture rafts and eelgrass may differ based on the type of
aquaculture raft and the environmental conditions of the eelgrass. Future studies of aquaculture
rafts and eelgrass in other areas impacted by historical tsunami events may improve our
understanding of these differences and the generalizability of the fragility functions.

**Acknowledgments**
We thank the Miyagi Prefecture Fisheries Cooperative Association (JF Miyagi) Ishinomaki Bay
branch for their information on the aquaculture rafts and Dr. Daisuke Sugawara (Museum of
Natural and Environmental History, Shizuoka) for his help in developing the bathymetry and
topography data. This study was funded by the Tokio Marine & Nichido Fire Insurance Co., Ltd.
through IRIDeS, Tohoku University, Willis Research Network (WRN) and JSPS Grant-in-Aid for
Young Scientists (B) "Applying developed fragility functions for the Global Tsunami Model
(GTM)" (grant no. 16K16371).

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

transport on seaweed bed dissipation in Shizugawa Bay, Miyagi Prefecture in the 2011 Great
East Japan Earthquake, Abstract of the 2015 Annual Seminar of Tohoku Disaster Science
Research (in Japanese).