# Peer review of "Developing fragility functions for aquaculture rafts and eelgrass in the case of the 2011 Great East Japan tsunami"

_Natural Hazards and Earth System Sciences, 2017_

## Referee Comment (RC1) · Anonymous Referee #1 · 3 Oct 2017

The manuscript contains relevant methodology and analysis based on data to develop the fragility curves for marine systems (aquaculture rafts and eelgrass). The results are new and applicable to further researches worldwide.

---

## Author Comment (AC1) · 4 Oct 2017

Thank you very much for your time in reviewing our manuscript and your comments. I wish that our results will be useful for future tsunami risk assessment in this field worldwide.

---

## Referee Comment (RC2) · Anonymous Referee #2 · 8 Nov 2017

Comments on the paper "Developing Fragility Functions for Aquaculture Rafts and Eelgrass in the case of the 2011 Great East Japan Tsunami" by Suppasri and co-authors

General Comments: The paper by Suppasri et al. addresses the tsunami damage on offshore systems through developing fragility functions for aquaculture rafts and eelgrass. To this end, the authors use the satellite data at two target areas damaged by the 2011 Great East Japan tsunami together with the numerical modelling of tsunami characteristics and the linear regression analysis. the phenomenon of tsunami waves overtopping the coastal protective structures through physical dam-break experiments. The authors conclude that the flow velocity is the main factor controlling the damage on

offshore marine plants regardless of the water level. While the MS is of great interest as it is the first attempt of establishing a method to quantify the damage on aquaculture rafts and eelgrass, I find that some points need to be carefully improved. Specific Comments: In terms of the MS structure and writing, the paper is of good quality; it is well structured and is easily readable. In terms of scientific content, in order to get the paper acceptable for publication in NHESS journal, the following comments should be carefully considered:

My main criticism of this work concerns the way the authors investigate the damage on the aquaculture rafts and therefore the damage classes attributed. In Sect 3.1 (p6, l187-188) they mention that "The remaining aquaculture rafts were classified as un-damaged, whereas the disappeared aquaculture rafts were classified as damaged", which I assume very simplistic for a quantitative damage assessment that require the consideration of different levels of damage (none, slight, moderate, high, and very high). This also applies to the Figs. 12 and 15, where the authors present their results of damage probability as function of flow velocity; but which kind of damage they refer to? A slight damage that can be easily repaired? Or a complete destruction? In other words, damaged and not damaged, is a kind of information that not help that much in tsunami recovery procedure. In the light of this comment, the authors are asked: first to provide a comprehensive classification of possible damages on the aquacul-ture rafts (damage classes definition), second to associate a specific damage class to each offshore marine system, and third, to develop fragility function for each damage class/level.

Minor suggestions regarding the Figures presentation: . Fig3: for Regions 1, 2, and 3 there are frames with geographical coordinates that do not much the limits of your grids. Please delete them . Fig4: In the legend specify which are the simulated and the measured tsunami heights . Fig 5: It is hard to distinguish between the wave height/flow velocity and some topographic elevations, both have yellow colour. Change the colour palette of one of them.

---

## Author Response (AR1)

**Reviewer no. 1**

Comments: The manuscript contains relevant methodology and analysis based on data to develop the fragility curves for marine systems (aquaculture rafts and eelgrass). The results are new and applicable to further researches worldwide.

Answers: Thank you very much for your time in reviewing our manuscript and your comments. We wish that our results will be useful for future tsunami risk assessment in this field worldwide.

**Reviewer no. 2**

Comments: General Comments: The paper by Suppasri et al. addresses the tsunami damage on offshore systems through developing fragility functions for aquaculture rafts and eelgrass. To this end, the authors use the satellite data at two target areas damaged by the 2011 Great East Japan tsunami together with the numerical modelling of tsunami characteristics and the linear regression analysis. the phenomenon of tsunami waves overtopping the coastal protective structures through physical dam-break experiments. The authors conclude that the flow velocity is the main factor controlling the damage on offshore marine plants regardless of the water level. While the MS is of great interest as it is the first attempt of establishing a method to quantify the damage on aquaculture rafts and eelgrass, I find that some points need to be carefully improved. Specific Comments: In terms of the MS structure and writing, the paper is of good quality; it is well structured and is easily readable. In terms of scientific content, in order to get the paper acceptable for publication in NHESS journal, the following comments should be carefully considered.

Answers: We highly appreciate the time that the reviewer spent in reviewing our manuscript. Their comments and suggestions are valuable, especially about the clarification of the damage definition and criteria used in our study. Please see below our answers, responses and corrections regarding to your comments in detail.

Comments: My main criticism of this work concerns the way the authors investigate the damage on the aquaculture rafts and therefore the damage classes attributed. In Sect 3.1 (p6, l187-188) they mention that "The remaining aquaculture rafts were classified as undamaged, whereas the disappeared aquaculture rafts were classified as undamaged, whereas the disappeared aquaculture rafts were classified as undamaged, whereas the disappeared aquaculture rafts were classified as undamaged, whereas the disappeared aquaculture rafts were classified as undamaged, whereas the disappeared aquaculture rafts were classified as damaged", which I assume very simplistic for a quantitative damage assessment that require the

consideration of different levels of damage (none, slight, moderate, high, and very high). This also applies to the Figs. 12 and 15, where the authors present their results of damage probability as function of flow velocity; but which kind of damage they refer to? A slight damage that can be easily repaired? Or a complete destruction? In other words, damaged and not damaged, is a kind of information that not help that much in tsunami recovery procedure. In the light of this comment, the authors are asked: first to provide a comprehensive classification of possible damages on the aquaculture rafts (damage classes definition), second to associate a specific damage class to each offshore marine system, and third, to develop fragility function for each damage class/level.

Answers: Thank you very much for pointing out this important issue. We totally agreed that the damage definition should be more clearly specified. In order to this, we have added more explanations mainly in section 3.1 as well as other related sections about the damage level defined by the Japan Fisheries Agency (JF) and the damage used in our study. The classification by JF has four levels (complete, major, moderate and minor damages). This classification is used for assessing the damage as well as the recovery process. Our study used satellite images before and after the tsunami which only allowed us to investigate the complete damage (wash away) from other damages or undamaged. According to your question, we have now clearly mentioned that the developed fragility curves are for the complete damage level in both the explanations and figures 12 and 15. Please see more detail below on our answers and responses.

1. To provide a comprehensive classification of possible damages on the aquaculture rafts (damage classes definition):

There are four damage levels defined by JF, namely, 1) complete damage (wash away), 2) major damage (70-100% damage), 3) moderate damage (30-70% damage) and 4) minor damage (less than 30% damage).

2. To associate a specific damage class to each offshore marine system:

As the pioneer study, we could only use the satellite images before and after the tsunami. Based on the quality of the image, we could only be classified whether the rafts were washed away (complete damage) or not. We are now working on collecting the actual data from the local fishery agencies but there are some difficulties as the damage data is related to personal information. We may be able to get with several document works or might not possible. Therefore, using the satellite images for the damage classification was only the best method we could you for now.

3. To develop fragility function for each damage class/level:

As mentioned above, this study could only able to develop the fragility functions for the complete damage level. In our future study, fragility functions for other damage levels will be certainly considered using the actual damage data if we can have access to such data.

Comments: Minor suggestions regarding the Figures presentation.

Fig3: for Regions 1, 2, and 3 there are frames with geographical coordinates that do not much the limits of your grids. Please delete them.

Fig4: In the legend specify which are the simulated and the measured tsunami heights.

Fig 5: It is hard to distinguish between the wave height/flow velocity and some topographic elevations, both have yellow color. Change the color palette of one of them.

Answers:

Fig. 3: We understood the point mentioned by the reviewer. However, our tsunami simulation was done using the nesting grid systems and it is important to show readers the coverage area of our simulation from the tsunami source.

Fig. 4: We have changed the wordings from "calculated" and "observed" to more understandable "simulated" and "measured" as your suggestion.

Fig. 5: We have modified the color bar according to your suggestion. Now color ranges of tsunami and topography are totally different.